# Scalable Approaches for a Theory of Many Minds

**Maximilian Puelma Touzel** [* 1 2]  **Amin Memarian** [* 1 2]  **Matthew Riemer** [1 2 3]  **Andrei Mircea Romascanu** [1 2]
**Andrew Williams** [1 2]  **Elin Ahlstrand** [4]  **Lucas Lehnert** [5]  **Rupali Bhati** [6]  **Guillaume Dumas** [7 2]  **Irina Rish** [1 2]

## Abstract

A major challenge as we move towards building agents for real-world problems, which could involve a massive number of human and/or machine agents, is that we must learn to reason about the behavior of these many other agents. In this paper, we consider the problem of scaling a predictive theory of mind (ToM) model to many interacting agents with a fixed computational budget. Motivated by the limited diversity of agent types, existing approaches to scalable ToM learn versatile single-agent representations for quickly adapting to new agents encountered sequentially. We consider the more general setting that many agents are observed in parallel and formulate the corresponding *Theory of Many Minds* (ToMM) problem of estimating the joint policy. We frame the scaling behavior of solutions in terms of parameter sharing schemes and in particular propose two parameter-free architectural features that endow models with the ability to exploit action correlations: encoding a multi-agent context, and decoding through an abstracted joint action space. The increased predictive capabilities that have come with foundation models have made it easier to imagine the possibility of using these models to make simulations that imitate the behavior of many agents within complex real-world systems. Performing these simulations in a general-purpose way would not only help make more capable agents, but it also could support applications in social science, political science, and economics.

---

[*]Equal contribution  [1]Department of Computer Science and Operations Research, University of Montreal, Montreal, Canada [2]Mila - Quebec AI institute, Montreal, Canada [3]IBM, New York, USA [4]Independent Researcher [5]University of Saskatchewan, Saskatoon, Canada [6]Northeastern University, Boston, USA [7]CHU Sainte-Justine Research Center, Department of Psychiatry, University of Montréal, Montreal, Canada. Correspondence to: Maximilian Puelma Touze <puelmatm@mila.ca>.

*Proceedings of the 41ˢᵗ International Conference on Machine Learning*, Vienna, Austria. PMLR 235, 2024. Copyright 2024 by the author(s).

## 1. The Problem

We adopt the Markov Decision Process formalism from standard multi-agent reinforcement learning. With state space $\mathcal{S}$ and action space for each agent $\mathcal{A}_i$ for $i = 1, \ldots, N+1$, let $T(s'|s, a_1, \ldots, a_{N+1})$ be the system's transition function with $s \in \mathcal{S}$ and $s' \in \mathcal{S}$ the current and next state, $a_i \in \mathcal{A}_i$ sampled from the $i$th agent's policy $\pi_i(a_i|s)$. From the perspective of agent $N+1$, the effective system transition function, $T_{N+1}$, is

$$T_{N+1}(s'|s, a_{N+1}) = \tag{1}$$
$$\sum_{a_1, \ldots, a_N} T(s'|s, a_1, \ldots, a_{N+1}) \cdot \underbrace{\pi_1(a_1|s) \cdots \pi_N(a_N|s)}_{\text{ToMM: estimation of } \pi(\boldsymbol{a}|s)} \ .$$

Even in decentralized and model-free settings, it is then necessary for agent $N+1$ to predict the actions of other agents to make optimal decisions and maximize rewards. For such an embedded agent, there is real utility in an efficient estimate of the joint policy of other agents. Even for an external agent tasked only with building a model of the $N$-agent system, the observations of states and actions can serve as the signal with which the joint policy can be learned. We term this the *Theory of Many Mind* (ToMM) problem. We leave further development of the embedded agent decision-making problem to future work, hereon omit mention of agent $N+1$, and focus on the ToMM problem of estimating joint policy $\boldsymbol{\pi}(\boldsymbol{a}|s)$ of an $N$-agent 'ground' system, where $N$ can be arbitrarily large so that abstraction becomes necessary where possible. Per notation: for every agent-specific variable $x$, we denote by boldface $\boldsymbol{x}$ the vector of that quantity over the $N$ agents, $\boldsymbol{x} = (x_1, \ldots, x_N)$. The ToMM problem centers on joint policy estimation from a $T$-sized dataset of states and joint actions $\mathcal{D}_T = \{(s_d, \boldsymbol{a}_d)\}_{d=1}^T$ obtained from the ground system by observing $N$ agents interacting together with some unknown environment and thus unknown system transitions. Market transactions are one real world example.

While agents' actions are conditionally independent of each other given the state and each's parameters, we assume the agents have learned their policies in the presence of each other so that action correlations can arise, certainly across states but even in a given state. To account for these ground system correlations despite not having access to the learning

process that generated them, the joint policy model used in estimation, $\pi_\Theta(\boldsymbol{a}|s)$ with model parameters $\Theta$, is allowed to exploit parameter sharing (or more generally mixing) schemes in either the case of deterministic or stochastic policies to achieve the same ends. As we will see, however, indexing individual agents' policies can become impossible. This is resolved by conditioning the policies on a to-be-determined single-agent context variable, $x_d$, derived from the dataset, in addition to conditioning on the state sample $s_d$.

We then maximize the model likelihood given the data (negative of the cross entropy loss),

$$\mathcal{L}(\Theta|\mathcal{D}_T) = \sum_{d=1}^{T} \sum_{i=1}^{N} \ln \pi_{\Theta_i}(a_{d,i}|s_d, x_{d,i}) , \quad (2)$$

where free parameters $\Theta$ determine all agent-specific parameters $\Theta_i$. In massively many-agent systems, $N$ is the bottleneck in computational cost. We are interested in scalable approaches to this problem, so we will investigate how quantities depend on $N$. For example, we focus on models whose size (*i.e.* number of parameters) scales weakly or not at all with $N$. There are also memory constraints. Given efficient parallelization schemes in modern architectures, especially for matrix operations, we are less concerned with how compute scales with $N$.

## 2. Existing Solutions

**Specific Setting** We consider a $S$-dimensional state space $\mathcal{S} = \mathbb{R}^S$ and for simplicity, assume agents share the same discrete action space $\mathcal{A}$ of size $A := |\mathcal{A}|$. We do not explicitly consider the case where agents observe each other. All such observations must be through $s$, and are limited since we assume $S$ cannot depend on $N$.

We choose a neural network-based encoder-decoder policy model architecture for a versatile policy model class that can capture the complexity of real-world agents. We denote the vector of latent variables across agents $\boldsymbol{z}$. An example of $\boldsymbol{z}$ is the private observations of the correlating device in a correlated equilibrium. The policy model we consider is then

$$\pi_{(\Theta_i^{\text{enc}}, \Theta_i^{\text{dec}})}(a_i, z_i|s, x_i) = p_{\Theta_i^{\text{dec}}}(a_i|z_i) p_{\Theta_i^{\text{enc}}}(z_i|s, x_i) \quad (3)$$

with encoder and decoder parameters for the policy denoted $\Theta_i^{\text{enc}}$ and $\Theta_i^{\text{dec}}$, respectively. We form the joint policy by the product of single-agent policies along with a parameter sharing scheme, $(\boldsymbol{\Theta}^{\text{enc}}, \boldsymbol{\Theta}^{\text{dec}}) = \mathcal{P}(\Theta)$ that maps the list of free parameters, $\Theta$, to all parameters, here collected on the encoding and decoding side, respectively. The joint policy

is then written,

$$\boldsymbol{\pi}_\Theta(\boldsymbol{a}, \boldsymbol{z}|s, \boldsymbol{x}) = \prod_{i=1}^{N} \pi_{(\Theta_i^{\text{enc}}, \Theta_i^{\text{dec}})}(a_i, z_i|s, x_i). \quad (4)$$

For simplicity, we assume deterministic encoder functions, $z_i = f_{\Theta_i^{\text{enc}}}(s, x_i)$, such that the conditioning on $s$ removes $\boldsymbol{z}$ from the random ensemble consistent with Equation (2).

### 2.1. Parameter sharing

The most direct means to keep the size of the joint policy model Equation (4) from growing with $N$ is to share parameters across agents. To motivate this, we first illustrate the benefit of parameter sharing in the many-agent regime by considering a Multi-Layer Perceptron (MLP) policy network model with the last layer interpreted as the decoder and analyze the model sizes under different sharing schemes: no sharing, only sharing the encoder parameters, and full sharing of both the encoder and the decoder. The first three rows of Table 1 list the model sizes of these three schemes, where we see the importance of sharing in achieving $\mathcal{O}(1)$ scaling with the number of agents. In Figure 1, we compare the capacity at fixed model size of these three schemes for a specific MLP. We see the same point.

Sharing requires that we provide an alternative source of conditioning the decoding policy on agents or (more efficiently) on agent types. Motivated by the fact that, through observation, agents are ultimately only distinguishable by the observed history of their actions, the learning setting for the shared MLP baseline augments the state of the current training sample $s_d$ with an $\mathcal{I}_d$-indexed state-action context block $x_{d,i} = \{(s_{d'}, a_{i,d'})\}_{d' \in \mathcal{I}_d}$ of length $|\mathcal{I}_d| = n_{\text{steps}}$ that serves as context (*e.g.* the last $n_{\text{steps}}$ state-action tuples). When $n_{\text{steps}}$ is set large enough, the encoder can learn to use the information in $x_{d,i}$ to distinguish agent $i$ from the rest. In such a case, the latent vector $z_i$ can both distinguish agents and, working with the decoder, learn to represent predictive features. Feeding it as input into a decoder policy network can thus effectively condition the decoder policy on the agent. To distinguish $N$ agents in general, the context size must increase as $N$ grows. For a sequence of $n_{\text{steps}}$, there are at most $A^{n_{\text{steps}}}$ possible sequences such that $n_{\text{steps}}$ must scale at least logarithmically, $n_{\text{steps}} \sim \Omega(\ln N)$. High amounts of positive correlation across agents can, however, bring this scaling up to $\mathcal{O}(N)$, impeding scalability.

### 2.2. Encoding with a shared sequence model

A strong baseline in state-of-the-art scalable single-agent ToM approaches is ToMnet (Rabinowitz et al., 2018). As with our sharing baseline, ToMnet achieves few-shot learning capability on novel agents by imitation learning a mapping from an agent's action history to a vector embedding

*Table 1.* Model size (parameter count) of considered architectures. Rows 1 to 3: 3 sharing schemes: no sharing, encoder-only sharing, and full sharing. $H$ is the number of hidden layers. Rows 4 and 5: Two modifications of the sharing baseline. The first is to a sequence model-based encoder that passes the context, one step at a time, through a sequence model of dimension $W_{\mathrm{SE}}$ (this avoids scaling by the context size; ToMnet is an instance of this class). The second is to a buffer attention-based decoder such that the decoder has no parameters (similar to Matching Networks(Vinyals et al., 2016)).

| Model name | Enc./Dec. sharing | free parameters, $\Theta$ | # for encoding | | # for decoding | |
|---|---|---|---|---|---|---|
| no sharing | no/no | $(\Theta^{\mathrm{enc}}, \Theta^{\mathrm{dec}})$ | $N(SW + HW^2)$ | $\mathcal{O}(N)$ | $NWA$ | $\mathcal{O}(N)$ |
| encoder-only | yes/no | $(\Theta^{\mathrm{enc}}, \Theta^{\mathrm{dec}})$ | $SW + HW^2$ | $\mathcal{O}(1)$ | $NWA$ | $\mathcal{O}(N)$ |
| sharing (baseline) | yes/yes | $(\Theta^{\mathrm{enc}}, \Theta^{\mathrm{dec}})$ | $(S + n_{\mathrm{steps}}(S + A))W + HW^2$ | $\Omega(\ln N)$ | $WA$ | $\mathcal{O}(1)$ |
| $\pm$Sequence Enc. | yes/yes | $(\Theta^{\mathrm{enc}}, \Theta^{\mathrm{dec}})$ | $(S + (S + A))W_{\mathrm{SE}} + W_{\mathrm{SE}}^2 + W_{\mathrm{SE}}W$ | $\mathcal{O}(1)$ | - | |
| $\pm$Buffer Att. Dec. | yes/yes | $(\Theta^{\mathrm{enc}}, \Theta^{\mathrm{dec}})$ | - | | 1 | $\mathcal{O}(1)$ |

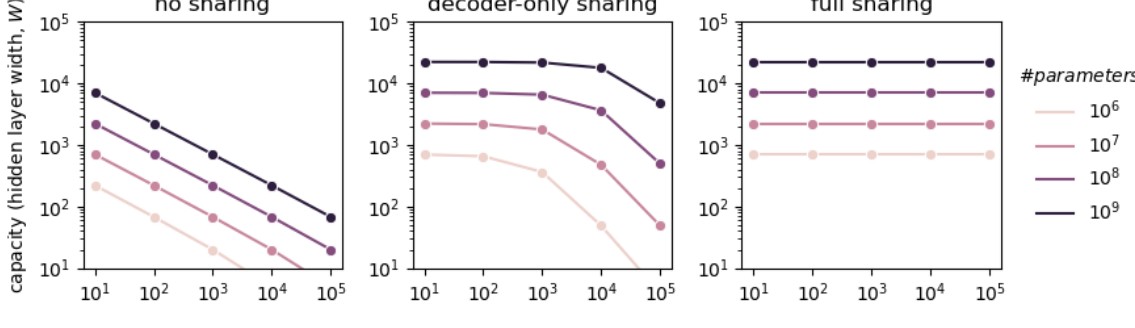

*Figure 1.* The scaling benefit of parameter sharing in joint policy models. Model capacity (hidden layer width, $W$) at fixed model size (number of parameters, $P$) can only be maintained in larger systems (large $N$) through parameter sharing. The first two architectures take the state, $s$, while the shared MLP takes, in addition, a state-action sequence. ($H = 2$, $A = 2$, $S = 8$, and $n_{\mathrm{steps}} = 16$.)

representation of agents. To improve over the baseline, ToMnet replaces the MLP encoder with a sequence model that processes the context in sequence before mapping to the latent, $z_i$. So long as the sequence model are able to represent the sequences uniquely, the shared sequence model encoder (SSE) architecture has two benefits. First, and crucially for scalable approaches, it removes the last remaining dependence on $N$ by removing the dependence on the sequence length $n_{\mathrm{steps}}$ of the context (see row 4 in Table 1). Second, the mixing properties of most sequence models vastly exceed that of MLPs such that, in principle, they can more efficiently capture dependencies across context iterates. While a notable advance, ToMnet was designed to learn from and predict the actions of only isolated single agents, encountered sequentially, both of which are unnecessarily restrictive assumptions.

## 3. ToMM solutions for the parallel observation regime

In contrast with the ToM setting considered in the ToMnet paper, observations and predictions of multiple agents can be made in parallel, and, in fact, this is both more realistic and useful when planning multi-agent systems. Relative to the ToM framing, this theory of many minds (ToMM) problem setting makes available additional opportunities for more efficient algorithm design. For example, correlated behaviour among agents could make learning more efficient by reducing the policy entropy and thus the sample complexity needed to estimate the policy to a desired level of accuracy. There are two natural architectural modifications that we here propose to exploit this reduced entropy.

### 3.1. Encoder-side: Incorporating interactions when sequence-processing the context

With the parallel processing of $N$ agents through the policy model, there arises the opportunity in SSEs to append to the sequence model's step function or to the entire sequence processing an interaction step in which the hidden states of different agents can affect each other. To avoid adding any $N$-dependent parameters, we investigate a simple multi-head self-attention step prominent in the transformer class of sequence models (*e.g.* in recurrent independent mechanisms (Goyal et al., 2021) and the decision transformer (Chen et al., 2021)). For $h \in \mathbb{R}^{W_{\mathrm{SE}}}$ the $N$-dimensional vector of

hidden states of the sequence model over all agents, and $\tilde{h}$ the hidden state output from the sequence model's step function, we add the attention-weighted interaction term,

$$h_{t+1} = \tilde{h}_{t+1} + \text{softmax}\left(\frac{\tilde{h}_{t+1}\tilde{h}_{t+1}^\top}{\sqrt{W_{\text{SE}}}}\right)\tilde{h}_{t+1} , \quad (5)$$

where the $\text{softmax}$ operation is applied row-wise. While this adds an $N^2$ operation, it is linear and thus efficiently parallelized. The processed sequence is then transformed into latent $z_i$ as before.

### 3.2. Decoder-side: Abstracting the joint action representation

In the spirit of the group mind construct from sociology, we are not bound to consider each agent an atom, and can construct a decoder representation that abstracts the $N$ agents by indexing joint actions. For this, we take inspiration from nearest neighbor classifiers that form a target prediction using state similarity-based attention over a sample memory buffer populated from training samples, *i.e.* the targets seen so far. A popular example is the matching network architecture (Vinyals et al., 2016) that incorporates a neural network state encoder and computes similarity in the space of latent variables. The natural application of the idea for our purpose is to replace the learned decoder with a dynamically-sized buffer of latents and joint actions, $B_b = ((\boldsymbol{z}_{b'}, \boldsymbol{a}_{b'}))_{b'=1}^{b-1}$, with $b$ incrementing with the processing of each novel training sample and to compute the joint action logits as a weighted combination of joint action targets seen so far ($N$ single-agent actions represented as one-hots), with weights computed as attention scores of the current latent $\boldsymbol{z}_b$ with each latent sample $\boldsymbol{z}_{b'}$ currently in the buffer. Denoting tensors of buffer latents and joint actions after $b$ training samples, $(B^z)_{b,i,j} \in \mathbb{R}^{(b-1)\times N \times W}$ and $(B^a)_{b,i,k} \in \mathbb{R}^{(b-1)\times N \times A}$, respectively, and denoting shared encoder, $f_\Theta$, the logit of the $k^{\text{th}}$ action probability of agent $i$, $\pi_\Theta(a_i = k|s, x_i)$, is

$$\text{softmax}\left(\frac{f_\Theta(s, x_i)(B^z)_{\cdot,i,\cdot}^\top}{\sqrt{N}}\right)(B^a)_{\cdot,i,k} . \quad (6)$$

While being parameter-free is certainly an advantage, we chose this attention-buffer decoder because of the way the structure of the buffer (and so that of the predictions) naturally admits action space abstraction. We can implement the representation of joint actions as either a factorized representation of single actions, $\boldsymbol{a}$, where each joint action is specified by $N$ integers (each encoded with $\log A$ bits). Alternatively, we can represent joint actions in an abstracted representation by specifying a single integer (encoded by $\log(A^N)$ bits) that indexes each of the $A^N$ possible joint actions. The two representations have the same total capacity ($N \log A$ vs. $\log A^N$) (same as the scaling of a PAC

bound on sample complexity of count-based action probability estimation; see Appendix A) but using the abstracted representation for the buffer has two benefits: first, the rate of adding samples begins at 1 (compared with a starting rate of $N$ for the factorized representation); second, in the case that actions are correlated such that the joint policy entropy is much reduced from the product of its marginals, the number of joint actions taken by the joint policy may be relatively quite small. Storing all training examples is memory intensive. However, sample (*e.g.* replay) buffers have been studied intensively in recent years, and there are now many approaches to making such buffers memory efficient.

## 4. Extension to multiple abstractions

The above is our exposition of *action space abstraction* through a monolithic abstraction of $N$ agents into a single group. The more realistic and thus interesting case is of an abstraction into multiple (soft or hard-assigned) groups. In fact, we selected the methods proposed here for this more general case of $M > 1$ groups, but as a presentation choice in this first paper, we decided to leave the explicit treatment of the general case for a follow-up paper. We nevertheless outline this extension below because it completes the description of the general ToMM problem (our larger goal).

In the $M$-group setting, the decoder has within-group parameter sharing, *i.e.* there are $M$ distinct encoders. As a result, the model size now scales with $M$. As for the buffer attention, it naturally factorizes into group action representations for which the action space abstraction discussed in this paper applies. A main feature of the ToMM problem with multiple groups is that agents must be assigned to groups. We call this *agent abstraction*. This assignment can be soft by design, or effectively soft by posterior inference over hard assignments. Naively, assignment adds $\mathcal{O}(N)$ parameters, *e.g.* via an $N \times M$ matrix of assignment probabilities.

Thus, a ToMM problem is, in fact, specified by not one but two abstraction subproblems: joint action space abstraction within groups and agent abstraction, *i.e.* assignment of agents to groups.

## 5. Summary

We provided two means by which versatile and scalable single-agent policy models can be augmented to efficiently learn from multiple agents in parallel. Our approach is a practically-minded, data-driven alternative to mean-field approaches to tackling many-agent systems. As fine-grained data on agent history becomes increasingly available, our formulation of the ToMM problem becomes increasingly relevant for traditional financial and consumer marketplaces as well as for efforts to capture the zeitgeist of political and social groups.

## Acknowledgements

We acknowledge the support from the Canada CIFAR AI Chair Program and the Canada Excellence Research Chairs (CERC) Program.

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

# A. PAC-bound derivation

We assume each step provides an *iid* sample of $(s, \boldsymbol{a})$ from an unknown distribution $p(s, \boldsymbol{a}) = \boldsymbol{\pi}(\boldsymbol{a}|s)p(s)$, where $p(s)$ is the stationary distribution over states. How large should the number of samples, $T$? As shown in the following derivation, we then have the desired estimation precision $\varepsilon > 0$, with probability at least $1 - \delta$ if $T$ satisfies

$$\frac{A_{\max}}{\varepsilon^2} \left( 4 - 16 \log \left( 1 - (1 - \delta)^{\frac{1}{N}} \right) \right) \leq T, \tag{7}$$

for $T \gg 1$, where $A_{\max}$ is the size of the largest single-agent action space. This lower bound grows linearly with $A_{\max}$ and logarithmically with $N$ (when $N$ is large) as more samples are needed to achieve the desired precision, $\varepsilon$.

When joining agents together in a joint action abstraction, the effective $N$ is lowered, but the effective $A_{\max}$ can increase, exponentially in the worst case ($A_{\max} = A^N$ for a single abstract agent). It is thus the tradeoff between the two that determines the optimal amount of abstraction in terms of the lowest lowerbound on the number of samples needed (See Figure 2). Take the case of $|\mathcal{A}_i| = 2$ ($a_i \in \{0, 1\}$) in which case the policy of agent $i$ is a Bernoulli distribution for each

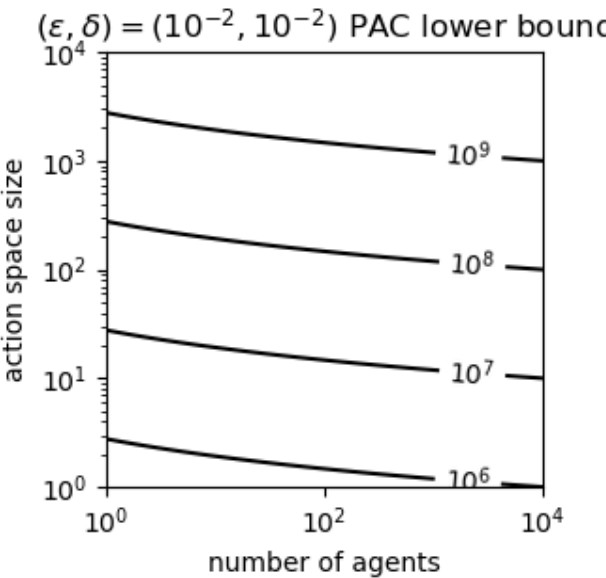

*Figure 2.* Contour lines of the PAC bound on sample complexity as a function of number of agents and largest action space size.

state. In this simple case, there is a known PAC bound for $\hat{p}_T(s, a_i = 1)$ assuming $s$ is fixed. It states that for any $\varepsilon > 0$,

$$\mathbb{P} \left( |\hat{p}_T(s, a_i = 1) - p(s, a_i = 1)| \geq \varepsilon \right) \leq \delta_i := 2e^{-2T\epsilon^2} . \tag{8}$$

Hence, the precision is achieved for all $N$ agents with probability at least $1 - \delta$ if $T$ satisfies

$$\frac{2}{\varepsilon^2} \left( \frac{1}{4} \log 2 - \frac{1}{4} \log \left( 1 - (1 - \delta)^{\frac{1}{N}} \right) \right) \leq T . \tag{9}$$

Our more general bound Equation (7) reduces to this established PAC bound when evaluated on $A_{\max} = 2$ (at least in scaling behaviour; the coefficients and constant term of the known result give a tighter bound). Our goal here is not to achieve a better bound, but to illustrate the generality of the form of scaling dependence.

One salient feature of Figure 2 is the relatively weak dependence on $N$ (*i.e.* the contours are almost horizontal). We interpret this as a result of the setting: each datum provides the $N$ actions taken by all $N$ agents. This is a rather optimistic, even unrealistic degree of system observability. Typically, only a fraction of a many-agent system is observable.

## A.1. Derivation

All sample-based quantities depend on the realization of $\mathcal{D}_T$ even though we will not denote this dependence explicitly. Let $m_T(s, \boldsymbol{a})$ denote the number of times the state-joint action tuple $(s, \boldsymbol{a})$ appears in $\mathcal{D}_T$. These state-joint action counts map through the given abstraction to counts of state-abstract action tuples for each abstract agent. Under the *iid* assumption and without prior information, the count distribution is multinomial for which the maximum likelihood estimate (MLE) of the probability given the observed sequence is simply the empirical frequency. We aim to obtain the convergence rates of this estimator onto the true probability.

We first present the baseline case of no abstraction and for only a single agent and its corresponding set of $|\mathcal{S}|(|\mathcal{A}| - 1)$ count variables, $m_T(s, a_i)$, for agent $i \in \{1, \ldots, N\}$. The MLE for $p(s, a_i)$ is $\hat{p}_T(s, a_i) = m_T(s, a_i) / \sum_{(s,a_i)} m_T(s, a_i)$. With an estimate of all $p(s, a_i)$ probabilities, we can form an estimate of agent $i$'s policy from the ratio estimator,

$$\hat{p}_T(a_i|s) := \hat{p}_T(s, a_i) \bigg/ \sum_{a_i \in \mathcal{A}} \hat{p}_T(s, a_i), \tag{10}$$

whose distribution propagates the uncertainty in $\hat{p}_T(s, a_i)$.

With the estimator defined, we can apply PAC theory to obtain its convergence. For example, take the general case of $N$ agents each having a distinct action space $\mathcal{A}_i$, but in an environment with only a single state, $s$, for simplicity. The standard PAC approach considers the joint event that estimated probability for each and every agent falls within $\varepsilon$. The probability of this event factorizes into the product of probabilities of each event individually,

$$\mathbb{P}\left( \bigwedge_{i=1}^{N} \left\{ ||\hat{p}_T(s, a_i) - p(s, a_i)|| < \varepsilon \right\} \bigg| s \right) = \prod_{i=1}^{N} \mathbb{P}\left( ||\hat{p}_T(s, a_i) - p(s, a_i)|| < \varepsilon \big| s \right) \tag{11}$$

$$= \prod_{i=1}^{N} \left[ 1 - \mathbb{P}\left( ||\hat{p}_T(s, a_i) - p(s, a_i)|| \geq \varepsilon \big| s \right) \right] \tag{12}$$

$$\geq \prod_{i=1}^{N} \left[ 1 - \delta_i \right], \tag{13}$$

where in the last line we have used an upper bound, $\delta_i$ on the probability that the estimate for agent $i$ deviates away from the actual value by at least $\varepsilon$. Using standard bound derivation techniques, we can use $\delta_i := \exp\left( -\frac{\varepsilon^2}{16|\mathcal{A}_i|} \cdot \frac{T^2}{T-1} + \frac{1}{4} \right)$. We can thus define the lower-bound $1 - \delta$ on the joint-event probability using the deviation $\delta := 1 - \left[ 1 - \delta_{\max} \right]^N$, where $\delta_{\max}$ is the deviation for the agent with the largest $|\mathcal{A}_i|$, denoted $A_{\max}$. The result is the bound shown as Equation (7).

