# OpenReview forum: "Scalable Approaches for a Theory of Many Minds"
_ICML.cc/2024/Workshop/Agentic_Markets — Agentic Markets @ ICML'24 Poster_

### Official Review · Reviewer_GR7f · 2024-06-13
**Interesting paper on modeling multi-agent behavior but lacking in further examples and results**

**Rating:** 6
**Confidence:** 2

**Review:**

The paper presents a new approach to representing the policies of multiple agents interacting with a fixed computational budget. Different from the approach of learning a versatile single agent representation, this work considers parallel observations of multiple agents. This is formulated as the Theory of Many Minds (ToMM) problem of joint policy estimation. The authors present an encoder and decoder architecture with various sharing schemes to both encode the shared context and also distinguish the actions of different agents.

The names of the authors are not anonymous. I am pointing this out, even though I am aware that the website says non-anonymous submissions are allowed. Other minor remarks:

- Equation 1 would be more succinct if the ToMM term was written using the product notations or as a separate equation altogether for $\pi(\textbf{a} |s)$.
- The architecture choice should be subject to more ablations and stronger arguments beyond the “versatile policy model class” claim.
- The mentioning of foundation models in the abstract comes across as forced since there is no additional mention or direct connection to the methods proposed or problems formalized on the paper.

Overall, the problem statement is very interesting and the architecture with the parameter sharing schemes seems very promising in conditioning group policies. The paper is a little difficult to read and is lacking in results other than the scaling benefit illustration. As is pointed out in section 4, the monolith representation is a very limited abstraction and a soft categorization would be more helpful as future work. This work is a good fit for the workshop as it can find application in a lot of market settings where the monolith representation could be closer to reality. A more general discussion of the modelling problem at hand would really improve the paper’s reachability.